# The Cardiotonic Steroid Marinobufagenin Is a Predictor of Increased Left Ventricular Mass in Obesity: The African-PREDICT Study

**DOI:** 10.3390/nu12103185

**Published:** 2020-10-18

**Authors:** Michél Strauss-Kruger, Ruan Kruger, Wayne Smith, Lebo F. Gafane-Matemane, Gontse Mokwatsi, Wen Wei, Olga V. Fedorova, Aletta E. Schutte

**Affiliations:** 1Hypertension in Africa Research Team (HART), North-West University, Potchefstroom 2520, South Africa; Straussmichel1@gmail.com (M.S.-K.); Ruan.kruger@g.nwu.ac.za (R.K.); wayne.smith@nwu.ac.za (W.S.); lebo.gafane@nwu.ac.za (L.F.G.-M.); 22368590@nwu.ac.za (G.M.); 2MRC Research Unit for Hypertension and Cardiovascular Disease, North-West University, Potchefstroom 2520, South Africa; 3National Institute on Aging, NIH, Baltimore, MD 212242, USA; wen.wei@nih.gov (W.W.); fedorovo@grc.nia.nih.gov (O.V.F.); 4School of Population Health, University of New South Wales, The George Institute for Global Health, Sydney 2052, Australia

**Keywords:** body mass index, cardiotonic steroids, left ventricular mass, marinobufagenin, obesity, dietary salt intake, young adults

## Abstract

The endogenous Na^+^/K^+^-ATPase inhibitor, marinobufagenin (MBG), strongly associates with salt intake and a greater left ventricular mass index (LVMi) in humans and was shown to promote cardiac fibrosis and hypertrophy in animals. The adverse effects of MBG on cardiac remodeling may be exacerbated with obesity, due to an increased sensitivity of Na^+^/K^+^-ATPase to MBG. This study determined whether MBG is related to the change in LVMi over time in adults with a body mass index (BMI) ≥30 kg/m^2^ (obese) and <30 kg/m^2^ (non-obese). The study followed 275 healthy participants (aged 20–30 years) from the African-Prospective study on the Early Detection and Identification of Cardiovascular disease and Hypertension (African-PREDICT) study over 4.5 years. At baseline, we measured 24 h urine MBG excretion. MBG levels were positively associated with salt intake. LVMi was determined by two-dimensional echocardiography at baseline and after >4.5 years. With multivariate adjusted analyses in obese adults (*N* = 56), we found a positive association of follow-up LVMi (Adjusted (Adj.) R^2^ = 0.35; Std. β = 0.311; *p* = 0.007) and percentage change in LVMi (Adj. R^2^ = 0.40; Std. β = 0.336; *p* = 0.003) with baseline MBG excretion. No association of LVMi (Adj. R^2^ = 0.37; *p* = 0.85) or percentage change in LVMi (Adj. R^2^ = 0.19; *p* = 0.68) with MBG excretion was evident in normal weight adults (*N* = 123). These findings suggest that obese adults may be more sensitive to the adverse cardiac effects of MBG and provide new insight into the potential role of dietary salt, by way of MBG, in the pathogenesis of cardiac remodeling in obese individuals.

## 1. Introduction

Obesity affects 671 million adults globally [1] and contributes significantly to the pathogenesis of cardiovascular disease (CVD) [2]. Obesity is associated with hypertension [3], left ventricular hypertrophy [4], and an overall greater risk of incident CVD [5]. Adipose tissue exerts an array of effects on the cardiovascular system through adipokines and low-grade inflammation [2] but also hemodynamically through volume loading [6]. Increased sodium retention observed in obese individuals [7] promotes extracellular volume expansion and concurrently causes a rightward shift in the renal function curve, ultimately elevating mean arterial pressure [6]. The latter mechanism is proposed to contribute to the development of hypertension [6] and cardiac remodeling [8] associated with obesity.

Obese individuals are also more likely to have poor dietary habits, such as higher caloric intake accompanied by excessive salt intake, which is the leading dietary risk factors associated with cardiovascular and all-cause mortality [9]. While the adverse effects of excess salt intake on the cardiovascular system is well known [10], more attention has been brought to the role of cardiotonic steroids, which are associated with increased salt intake in the development of CVD. This includes the biomarker marinobufagenin (MBG) [11], which is synthesized and secreted by the adrenal cortex in response to sodium loading [12]. Indeed, our group has previously demonstrated a strong positive correlation between MBG and estimated salt intake in young healthy adults [13].

MBG is an endogenous sodium-potassium adenosine triphosphatase (Na^+^/K^+^-ATPase) inhibitor that primarily promotes natriuresis in response to volume loading, acting as a compensatory mechanism to lower blood pressure [14]. However, during sustained periods of high salt intake, MBG levels continue to increase so that it ultimately evokes a pathophysiological response in the cardiovascular system. The latter occurs via the MBG inhibition of cardiovascular Na^+^/K^+^-ATPase [14]. Elevated MBG promotes cardiac fibrosis and hypertrophy in animals by way of the Na^+^/K^+^-ATPase-Src and/or Na^+^/K^+^-ATPase-SMAD-transforming growth factor beta (TGFβ) signaling cascades [15,16,17], which may involve oxidative stress initiated via Na^+^/K^+^-ATPase [18]. In humans with excessive MBG excretion, an independent cross-sectional association of MBG with increased left ventricular mass index (LVMi) [19] was demonstrated, but it is unknown whether obesity would further exacerbate this relationship.

In animal studies, obese rats were previously shown to have attenuated cardiac Na^+^/K^+^-ATPase expression when compared to lean controls [20]. In non-obese Na/K-ATPase α1 heterozygote knock-out mice (α1^+/−^) (displaying suppressed Na^+^/K^+^-ATPase expression), the effects of MBG were potentiated as shown by increased myocyte apoptosis and left ventricular dilation compared to wild-type mice in response to MBG infusion [21]. This may suggest that an attenuated expression of cardiomyocyte Na^+^/K^+^-ATPase with obesity [20] could increase the sensitivity of cardiac tissue to the pathological effects of MBG [22]. Therefore, it is likely that elevated MBG in obese individuals with a high habitual dietary salt intake may contribute to early cardiac remodeling. The female sex hormone estradiol has been shown to restore Na^+^/K^+^-ATPase activity in obese male Wistar rats, and it may play a cardioprotective role with obesity [20].

The aim of this study was to determine whether MBG is related to changes in LVMi over time in obese compared to non-obese individuals, and whether MBG predicts an increase in LVMi over time. The 4.5-year baseline and follow-up data of 275 young adults, aged 20–30 years at baseline, with no previous history of diagnosed CVD was analyzed. We hypothesize that obese participants will demonstrate an adverse association of MBG with follow-up LVMi as well as with the change in LVMi over 4.5 years. These associations were not expected to be seen in participants with a healthy body composition.

## 2. Results

Table 1 presents the basic characteristics of 275 participants at baseline (mean age 25.4 ± 3.16) and follow-up (mean age 30.0 ± 3.21), with an even distribution between sex and black and white ethnic groups (black: 50.2% and men 45.5%). The median time from baseline to follow-up was 1639 days (4.49 years). There was a marked increase in the body weight, body mass index (BMI), and waist/height ratio (WHtR) (all *p* < 0.001) from baseline to follow-up. In addition, there was an increase in the number of participants who were classified as obese using BMI (*N*
_baseline_ = 60 to *N*
_follow-up_ = 82) or a composite obesity score of BMI, waist circumference (WC), and WHtR criteria (*N*
_baseline_ = 55 to *N*
_follow-up_ = 74) (*p* < 0.001). Participants’ systolic blood pressure (SBP) (*p* < 0.001) and left ventricular mass index (LVMi, *p* < 0.001) increased significantly from baseline to follow-up, while the end diastolic volume index (EDVi) and stroke volume index (SVi) decreased (*p* < 0.001). There were 26 obese participants at baseline that had masked hypertension. When comparing the baseline characteristics of participants followed up in this sub-study (*N* = 275) with those not yet followed up (*N* = 927), the ethnic and sex distributions, blood pressure (BP), 24 h MBG excretion, and estimated salt intake were similar (Appendix A). However, the sub-group included in this study had higher BMI (0.99 kg/m^2^ mean difference) but lower LVMi (2.93 kg/m^2^ mean difference).

Baseline and follow-up characteristics of participants were compared, with participants stratified as non-obese (BMI < 30 kg/m^2^) and obese (BMI ≥ 30 kg/m^2^) (Appendix A). In both non-obese (*p* < 0.001) and obese (*p* = 0.029) participants, clinic SBP decreased significantly from baseline to follow-up, although central SBP (cSBP) increased only in obese adults (mean diff. 2.58 mmHg, 95%CI 0.17; 4.99) (*p* = 0.037). Non-obese and obese adults had an increase in LVMi and a decrease in EDVi and SVi (*p* < 0.001). Still, obese adults had a significantly greater increase in LVMi when compared to non-obese adults (mean diff. 5.81 g/m^2^, 95% CI 3.67; 7.95 vs. mean diff. 11.3 g/m^2^, 95%CI 6.43; 16.2; *p* = 0.024).

When comparing the LVMi of participants within different BMI categories at baseline, only underweight adults had a significantly smaller LVMi when compared to obese participants (*p* = 0.033). However, obese adults had significantly greater EDVi (71.8 mL/m) when compared to underweight (57.4 mL/m), normal weight (60.8 mL/m), and overweight participants (64.7 mL/m) at baseline (*p* < 0.001). At follow-up, we found that participants who were obese at baseline had a significantly greater percentage change in LVMi over time as well as follow-up LVMi when compared to participants with a normal BMI (*p* = 0.001), when adjusting for sex, ethnicity, age, and baseline LVMi (Figure 1A,B). However, follow-up EDVi in obese adults (61.4 mL/m) was similar to that of normal weight (59.6 mL/m) (*p* = 1.00) and overweight participants (61.7 mL/m) (*p* = 1.00), when adjusting for sex, ethnicity, age, and baseline EDVi. There were no significant differences in the estimated salt intake of obese adults (8.17 g/day) when compared to overweight (7.93 g/day) (*p* = 0.77), normal weight (7.64 g/day) (*p* = 0.45), or underweight participants (6.04 g/day) (*p* = 0.070). Despite no significant difference in the baseline estimated salt intake of participants between different BMI categories (*p* = 0.32) (Figure 1C), underweight adults had a lower MBG excretion when compared to overweight (*p* = 0.037) or obese adults (*p* = 0.024) (Figure 1D). MBG correlated positively with estimated salt intake (underweight: r = 0.494, *p* = 0.061; normal weight: r = 0.553, *p* < 0.001; overweight: r = 0.514, *p* < 0.001; obese: r = 0.470, *p* < 0.001).

### 2.1. Pearson, Partial, and Multiple Regression Analyses

Pearson correlations were performed between follow-up LVMi as well as the percentage change in LVMi with MBG excretion within non-obese and obese participants at baseline (Figure 2A,B). In non-obese adults, a positive correlation was found between follow-up LVMi and MBG excretion at baseline (r = 0.166; *p* = 0.015) and a negative correlation between percentage change in LVMi and MBG excretion (r = −0.139; *p* = 0.042), but these relationships lost significance with multivariate adjusted analysis (*p* > 0.05). In addition, when the non-obese group was additionally stratified as normal weight and overweight (Appendix A), there was no association between LVMi or the percentage change in LVMi and MBG excretion (all *p* > 0.05). However, in obese adults, follow-up LVMi (r = 0.392; *p* = 0.002), as well as the percentage change in LVMi (r = 0.352; *p* = 0.006) correlated positively with baseline MBG excretion. The association of LVMi (Adj. R^2^ = 0.35; Std. β = 0.311: *p* = 0.007) and percentage change in LVMi (Adj. R^2^ = 0.40; Std. β = 0.336: *p* = 0.003) with MBG excretion in obese participants remained significant after multivariate adjusted analyses (Table 2). The present results remained robust when repeating analyses in a sub-group (*N* = 51) with stringently defined obesity based on three composite criteria (BMI ≥ 30 kg/m^2^ and WC > 94 cm for white men; >81.2 cm for black men; >80 cm for white women; and >81 cm for black women and WHtR >0.5; LVMi: Adj. R^2^ = 0.36; Std. β = 0.310: *p* = 0.009; percentage change in LVMi: Adj. R^2^ = 0.37; Std. β = 0.350: *p* = 0.003; see Appendix A).

### 2.2. Sensitivity Analyses

#### 2.2.1. Estimated Salt Intake

We additionally adjusted for estimated salt intake to determine whether salt intake influences the relationship between LVMi and MBG excretion. The relationship with both LVMi and percentage change in LVMi remained robust in obese participants (Adj. R^2^ = 0.35; Std. β = 0.311: *p* = 0.008; Adj. R^2^ = 0.40; Std. β = 0.337: *p* = 0.003; see Table 2). When performing backward stepwise regression analysis with salt as the main independent variable (Appendix A), there was no association between estimated salt intake and LVMi.

#### 2.2.2. Estradiol

Estradiol was previously reported to increase Na^+^/K^+^-ATPase activity in obese animals [20], and it is suggested to play a cardiac protective role in obesity [22]. As expected, women from this study had higher levels of estradiol when compared to men (54.3 pg/mL vs. 37.0 pg/mL; *p* < 0.001; see Appendix A). Pearson correlations were performed to demonstrate the correlation of estradiol and the percentage change in LVMi in men and women (Appendix A). When performing multiple regression analyses with the additional adjustment for estradiol (Table 2), LVMi was negatively associated with estradiol in obese adults (Adj. R^2^ = 0.47; Std. β = −0.220: *p* = 0.045). Nonetheless, the positive relationship of LVMi and MBG excretion remained robust (Adj. R^2^ = 0.47; Std. β = 0.305: *p* = 0.007). In the non-obese group, the association of LVMi and percentage change in LVMi with MBG was not significant (*p* > 0.05). When analyses were repeated in normal weight and overweight adults, respectively, a significant negative association between MBG and percentage change in LVMi was evident in overweight adults (Adj. R^2^ = 0.32; Std. β = −0.221: *p* = 0.035; Appendix A).

## 3. Discussion

This is the first study to our knowledge that investigated the relationship of the cardiotonic steroid, MBG, with longitudinal cardiovascular data in a young human population with no previous history of CVD. The main finding from the present study was that baseline MBG excretion levels were associated with increased LVMi over 4.5 years in obese but not in normal weight individuals. These results support the hypothesis that the adverse effects of MBG in cardiac remodeling may be potentiated in obese adults, despite having similar MBG excretion levels when compared to normal weight adults.

A positive association of follow-up LVMi with MBG excretion in obese but not underweight, normal weight, or overweight adults supports the notion that an increased sensitivity of cardiotonic steroids may play a role in the pathogenesis of CVD in individuals with obesity [22]. A recent review by Obradovic et al. highlighted the potential adverse role of reduced Na^+^/K^+^-ATPase in the development of CVD with obesity [22], which is associated with the downregulation of Na^+^/K^+^-ATPase in animals [20] and humans [25]. The attenuated levels of Na^+^/K^+^-ATPase likely contribute to the deleterious effect of MBG on the heart. Indeed, non-obese α1-Na^+^/K^+^-ATPase knockout mice (α1^+/−^) with lower Na^+^/K^+^-ATPase levels (−38% lower Na^+^/K^+^-ATPase) were shown to have an increased myocardial Na^+^/K^+^-ATPase sensitivity to MBG infusion when compared to wild-type mice [21]. It is via the inhibition of cardiac Na^+^/K^+^-ATPase and the concurrent downward Src signaling cascade that MBG promotes cardiac remodeling [15,16]. Similarly, in the Dahl-S model of salt-sensitive hypertension, MBG-activated cardiovascular TGF-β pro-fibrotic signaling via Na^+^/K^+^-ATPase resulted in an increase in LVMi while also promoting the development of fibrosis and cardiac remodeling [17].

In contrast with our findings in obese adults, a negative association was observed between percentage change in LVMi and MBG excretion in overweight adults when including estradiol into the model. Estradiol can activate Na/K-ATPase in the cardiovascular system [20,26] and counterbalance an inhibitory effect of MBG on the cardiovascular Na/K-ATPase enzymatic activity. In addition, estradiol may exhibit its cardiac protective effect by means of attenuated vasoconstriction and tissue fibrosis by binding to other estrogen receptors (ER), including, ER-α, ER-β, and GPR−30 [27]. It is also known that estradiol, i.e., estrogen, can activate atrial natriuretic peptide (ANP) receptors and increase ANP levels [28,29]. It was demonstrated that ANP can counterbalance the pro-fibrotic and pro-hypertensive effects of MBG in the cardiovascular system and exhibit a synergistic effect with MBG on renal natriuretic function [30]. It might only be with the shift from an overweight to an obese phenotype that this natriuretic function of MBG is overridden by the pathophysiological effects thereof. The beneficial effect of estradiol in association with MBG in subjects with normal weight and moderately overweight subjects is an important observation and will merit further investigation. It is likely that in the obese subjects, the positive impact of estradiol is similar to the observation made in age-associated salt sensitivity in the animal model [30]. In addition, the increased sensitivity of Na^+^/K^+^-ATPase to MBG can underlie the pathophysiological function of MBG, as it was previously demonstrated in Dahl-S rats, which exhibited an increased sensitivity of the cardiovascular Na^+^/K^+^-ATPase to MBG [31,32]. Acute salt loading of the normotensive Sprague–Dawley rats and salt-sensitive Dahl-S rats was accompanied by a similar stimulation of MBG in both strains and by the inhibition of an aortic Na pump in Dahl-S rats only [32], which was likely due to the higher sensitivity of cardiovascular Na^+^/K^+^-ATPase to cardiotonic steroids in Dahl-S rats. Dahl-S rats on a high salt intake also developed left ventricular hypertrophy, which was accompanied by an increased sensitivity of cardiac Na^+^/K^+^-ATPase to MBG [31]. An increased sensitivity of cardiovascular Na^+^/K^+^-ATPase to the endogenous inhibitors may contribute to the exaggerated effect of MBG on the Na^+^/K^+^-ATPase-initiated down-stream pro-fibrotic signaling [15,17].

Taken together, the activation of the Na^+^/K^+^-ATPase-Src signaling cascade may be promoted with obesity and contribute to early cardiac remodeling. The inhibition of cardiac Na^+^/K^+^-ATPase as a result of MBG is associated with oxidative stress [33], cardiac myocyte apoptosis [21], collagen synthesis [15,17], and cardiac myocyte hypertrophy [17], which are all factors of overall structural cardiac changes. Meanwhile, the endocytosis of Na^+^/K^+^-ATPase with obesity is proposed to play an important role in promoting a pro-inflammatory environment [34]; further studies are needed to investigate the role of MBG on macrophage adhesion and cytokines activation in the cardiac remodeling of obese individuals.

While young obese adults had a similar LVMi to normal weight or overweight adults at baseline, the greater LVMi at follow-up in obese participants of this study, when compared to underweight and normal weight adults, is in accordance with previous reports [4]. In addition, the greater EDVi observed in obese adults at baseline suggests a volume-loading phenotype in this population associated with obesity [35]. Since this study made use of LVM indexed to body surface area (BSA), the present findings are independent of cardiac structural changes as a result of increased body weight with obesity [36]. Considering the young age of this study population, the findings of the present study highlight the importance of creating a greater awareness regarding the substantial impact of modifiable risk factors, such as obesity and diet, on cardiovascular health. Indeed, the estimated salt intake of this study population (7.73 g/day) exceeded the daily recommendation of the World Health Organization (5 g/day) [37]. In the present study, behavioral lifestyle choices such as high salt intake from a young age in the obese condition is already associated with changes in cardiac structure.

In a recent review by He et al., the authors highlighted the diverse mechanisms whereby a high dietary salt intake may contribute to a greater risk of CVD [10]. Eminently, of the 10 million CVD-related deaths attributed to dietary risk factors in 2017, a high dietary salt intake was the leading risk factor [9]. However, excessive dietary salt intake may also indirectly contribute to early CVD development as a result of elevated MBG [11], which is strongly related to estimated salt intake [13]. MBG was firstly described as a natriuretic hormone; it increased with sodium-induced volume loading [38]. Subsequent investigations into the pathophysiological role of MBG have yielded strong evidence of an adverse effect of excessive MBG on the cardiovascular system. In animal and experimental studies, MBG was shown to promote cardiac myocyte apoptosis [21], cardiovascular fibrosis, and cardiac hypertrophy [15,16,17,39]. While in humans, MBG was found to be associated with autonomic activity [40], microvascular dysfunction [41], arterial stiffness [13,42], and increased LVMi [19]. Given that obesity is known to be associated with left ventricular hypertrophy [4], these findings may help reiterate the importance of lowering salt intake in obese individuals who are already more susceptible to early CVD development due to a harmful cardiometabolic profile.

In animals [20,25] and humans [25], hyperinsulinemia, which is associated with obesity, has been implicated in the downregulation of Na^+^/K^+^-ATPase. Additionally, insulin resistance in Dahl-S animals consuming a high salt intake [43] may increase the sensitivity of Na^+^/K^+^-ATPase to CTS. Obesity is one of the facets of metabolic syndrome, and it is often accompanied by insulin resistance [2]. This creates an “ideal” condition for MBG, which can activate the vicious pro-fibrotic circle without being elevated to a pathologic level. Therefore, obese individuals may be predisposed to early cardiac remodeling due to insulin resistance, which sensitizes cardiovascular Na^+^/K^+^-ATPase to CTS. The estimation of the insulin sensitivity and its association with cardiovascular parameters and cardiovascular markers in young obese and non-obese individuals merits future investigation. Furthermore, a high-fat diet has also been shown to stimulate MBG synthesis in hyperlipidemic states [34]. Further studies are also necessary to determine the role of a habitual high dietary intake of sugar and trans fats in the adverse effects of MBG on the heart and vasculature of obese adults.

## 4. Strengths and Limitations

A major strength of the present study was the inclusion of longitudinal data of young black and white adults, free of CVD at baseline with complete baseline data on 24-h MBG and sodium excretion. However, the study included volunteers without random selection. The main limitation of this study was the small group of obese adults included in the analyses. Given that the present study only included *N* = 275 from the larger African-PREDICT study population (*N* = 1202), it is possible that selection bias may have contributed to the findings. The follow-up studies, which will be performed in a larger sample to substantiate these results, will merit future investigation. The small sample size was a limitation factor in performing sex-specific analyses, which is recommended for future studies.

## 5. Materials and Methods

### 5.1. Study Design and Methodology

This study included the data of the first 275 (50% black; 44% men) participants from the African-Prospective study on the Early Detection and Identification of Cardiovascular disease and Hypertension (African-PREDICT) with baseline MBG excretion and follow-up LVMi data.

The African-PREDICT study is a longitudinal study that aims to track and monitor the cardiovascular health profile of young black and white adults. Details on the study design, recruitment, and baseline measurements were previously published [44]. Participant recruitment for the study started in 2012 and was conducted in communities living in proximity to the Potchefstroom area in the northwest province of South Africa. Apparently healthy volunteers were screened at baseline for eligibility to be included into the study based on the following criteria: black or white ethnicity; aged 20–30 years; clinic blood pressure (BP) < 140/90; HIV uninfected; had no previous diagnosis of chronic illness (self-reported) or using any chronic medications; not pregnant or lactating.

Ultimately, *N* = 1202 (606 black and 596 white) eligible participants were enrolled in the baseline phase of the African-PREDICT study (February 2013 to November 2017). Follow-up data collection started in February 2018 and remains ongoing. This analysis made use of the follow-up data collected between 2018 and 2019. At the point of statistical analyses for this study, 430 participants had been successfully contacted and participated in follow-up measurements. Of the 430 participants who had taken part in follow-up measurements, we excluded 88 participants with missing MBG data and a further 67 with missing follow-up echocardiography data. Therefore, this study analyzed the data of 275 participants. Baseline and follow-up data collection were performed under controlled conditions, using good clinical practice at the Hypertension Research and Training Clinic at North-West University.

Written informed consent was obtained from all participants at baseline and follow-up, and all procedures adhered to Institutional Guidelines and the Declaration of Helsinki of 1975, which were revised in 2013. The African-PREDICT study was approved by the Health Research Ethics Committee of the North-West University, South Africa (NWU-00001-12-A1). The study is registered at ClinicalTrials.gov (identifier: NCT03292094). Assignable to the informed consent given by participants, the data for this study, which is centrally managed by the data manager using REDCap, can be obtained by means of the necessary arrangements with Prof Aletta E Schutte or Prof Carina Mels from the Hypertension in Africa Research Team (HART) [44]. Potential collaborators are invited to contact the principal investigator of African-PREDICT for further information.

### 5.2. Questionnaire and Anthropometric Data

Participants completed a General Health and Demographic Questionnaire at baseline and follow-up to obtain detailed information (self-reported) on demographics (age, sex, ethnicity, and socio-economic status) and medical history (medication use).

Anthropometric measurements including weight (kg; SECA 813 Electronic Scales), height (m; SECA 213 Portable Stadiometer; SECA, Hamburg, Germany), and WC (cm) (Lufkin Steel Anthropometric Tape; W606PM; Lufkin, Apex, NC, USA) were measured in triplicate, according to the guidelines of the International Society for the Advancement of Kinanthropometry. We subsequently calculated the BMI (weight (kg)/height (m^2^)), BSA (√height (cm)×weight (kg)/3600), and WHtR of participants. Participants were classified as underweight (BMI < 18.5 kg/m^2^), normal weight (BMI 18.5–24.9 kg/m^2^), overweight (25–29.9 kg/m^2^), or obese (BMI ≥ 30 kg/m^2^) according to baseline BMI [23]. As a sensitivity measure, we additionally identified participants who would be classified as obese based on baseline BMI > 30 kg/m^2^, WC (WC > 94 cm for white men; >81.2 cm for black men; >80 cm for white women, and >81 cm for black women) [23,24] and WHtR cut points (>0.5) [45] (i.e., participants who met all three criteria for obesity—BMI, WC, and WHtR—were classified as being obese).

### 5.3. Cardiovascular Measurements

The Dinamap Procare 100 Vital Signs Monitor (GE Medical Systems, Milwaukee, WI, USA) was used to measure SBP and diastolic (DBP) BP with an appropriately sized brachial blood pressure cuff. Participants were seated with their arm rested at heart level. Measurements were performed in duplicate on the left and right arm (with a five-minute resting period between measures). The mean SBP and DBP from the right arm was calculated [46]. Hypertension was defined as SBP ≥ 140 mmHg and or DBP ≥ 90 mmHg [47]. We made use of the SphygmoCor XCEL device (AtCor Medical Pty. Ltd., Sydney, Australia) to measure cSBP in duplicate by means of pulse wave analysis, while participants lay rested in a supine position with an appropriately sized brachial BP cuff fitted to the right upper arm. cSBP is determined using a general transfer function.

Two-dimensional echocardiography was done using the General Electric Healthcare Vivid E9 device (GE Vingmed Ultrasound A/S, Horten, Norway), a 2.5 to 3.5 MHz transducer, and a single electrocardiogram (ECG) lead. A medical clinical technician performed echocardiographic imaging according to standardized procedures outlined by the American Society of Echocardiography [36]. Left ventricular mass was calculated from dimensions using the Devereaux “cube” formula (LVM = 0.8 × 1.04 ((IVSd + LVIDd + PWTd)^3^ − LVIDd^3^) + 0.6 g), where IVS is the interventricular septum at end diastole, LVID is the LV internal diameter at end diastole, and PWT is the inferolateral wall thickness at end diastole. Left ventricular mass and LV volumes were determined using the biplane method. Since LVM is strongly influenced by body size, LVMi was indexed to BSA to minimize obesity-related changes in LVM. Reporting LVM indexed to BSA is recommended by the American Society of Echocardiography [36]. Stroke volume was indexed for height to the power 2.04 (SVi) and end diastolic volume (EDV) (calculated using the Teichholz formula) was indexed to height.

### 5.4. Biochemical Sampling and Biochemical Analyses

Participants were instructed to refrain from eating or drinking (except water) from at least 22:00 the night before measurements took place. A trained research nurse performed early morning biological sampling (before 09:30) using a sterile winged infusion set and syringes. Samples were immediately moved to an onsite laboratory, centrifuged, and aliquoted into cryovials. A detailed description on sample preparation was published elsewhere [19]. All biological samples were stored in onsite bio freezers at −80 °C.

Then, 24 h urine sampling was performed in accordance with protocols of the Pan American Health Organization/World Health Organization (PAHO/WHO), with participants being given instructions on how to ensure accurate collection [48]. All participants had a urinary volume >300 mL. Baseline MBG was analyzed from 24 h urine samples using a solid-phase Dissociation-Enhanced Lanthanide Fluorescent Immunoassay, which was based on a 4G4 anti-MBG mouse monoclonal antibody described in detail by Fedorova et al. [49]. Twenty-four-hour urinary sodium excretion was measured (Cobas Integra 400plus, Roche, Basel Switzerland) to calculate estimated salt intake. Estimated glomerular filtration rate was calculated using the Chronic Kidney Disease Epidemiology Collaboration Equation (CKD-EPI) no race equation [50].

From serum samples, total cholesterol, high-density lipoprotein cholesterol (HDL-C), low-density lipoprotein cholesterol (LDL-C), creatinine, C-reactive protein (CRP), and γ-glutamyl transferase (GGT) were measured (Cobas Integra 400plus, Roche, Basel Switzerland). Glucose was determined from sodium fluoride plasma (Cobas Integra 400plus, Roche, Switzerland).

### 5.5. Statistical Analyses

Statistical analyses were performed using SPSS version 26 (IBM; Armonk, New York, NY, USA) and created figures using GraphPad Prism version 5.0 (GraphPad Software Inc., La Jolla, CA, USA). Data following a normal distribution were presented as the arithmetic mean ± standard deviation, and variables with a non-Gaussian distribution were logarithmically transformed and presented as geometric mean (5th and 95th percentiles). Differences in the basic characteristics of participants from baseline to follow-up were determined using a paired t-test or Wilcoxon Signed-Ranks Test for parametric or non-parametric continuous variables, and McNemar tests for categorical variables. Baseline estimated salt intake and MBG excretion of participants were compared within different BMI statuses (underweight, normal weight, overweight, and obese) using ANCOVA, adjusting for sex, ethnicity, and age. Scatterplots demonstrating Pearson correlations between LVMi as well as the percentage change in LVMi overtime and MBG excretion are shown for different BMI statuses. The relationship of LVMi and the percentage change LVMi with MBG excretion was further explored using backward stepwise multiple regression analyses, considering the limited number of participants who were classified as obese and having follow-up LVMi data (*N* = 56). Backward stepwise regression models were adjusted for sex, ethnicity, age, clinic SBP, eGFR, glucose, HDL-C, CRP, GGT, and baseline LVMi. Sensitivity analyses were performed for estimated salt intake and estradiol to determine whether the latter had an influence on the relationship between MBG and LVMi. Backward stepwise regression analyses were repeated, whereby obesity was stringently defined as not only having a BMI >30 kg/m^2^ but also according to WC (WC > 94 cm for white men; >81.2 cm for black men; >80 cm for white women; and >81 cm for black women) [23,24] and WHtR cut points (>0.5) [45] (i.e., participants who met all three criteria for obesity—BMI, WC, and WHtR—were classified as being obese *N* = 51).

## 6. Conclusions

The findings from the present study indicate that obese adults may be more vulnerable to the adverse cardiac effects of excessive salt intake by means of increased MBG sensitivity. This was demonstrated by an independent positive association between follow-up LVMi and MBG excretion in obese adults only. An increased sensitivity to MBG may exacerbate the vulnerability of obese adults to the harmful effects of excessive salt intake on cardiac remodeling, besides volume loading. These findings encourage cardioprotective strategies such as the targeting of modifiable risk factors (a healthy diet, low salt intake, weight reduction).

## Figures and Tables

**Figure 1 nutrients-12-03185-f001:**
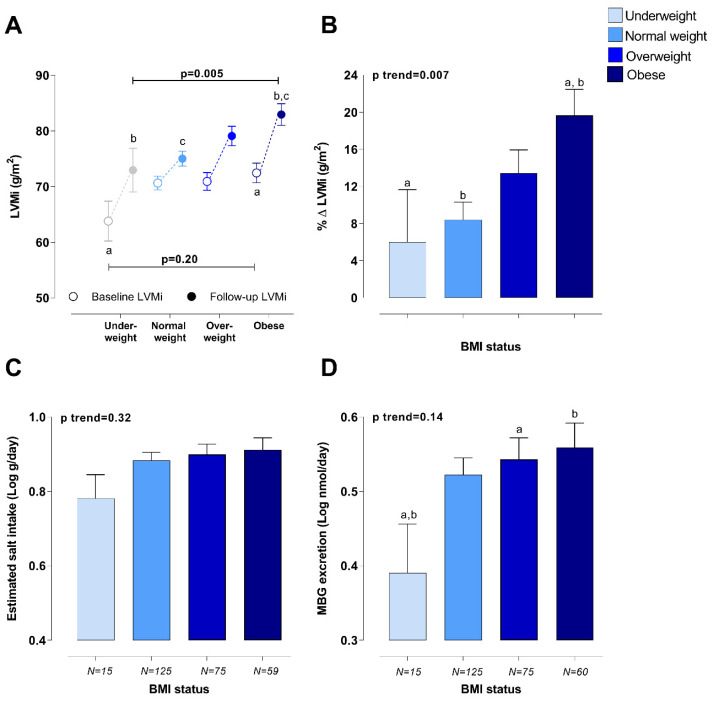
Comparison of (**A**) LVMi, (**B**) percentage change in LVMi, (**C**) baseline estimated salt intake, and (**D**) baseline MBG excretion levels of participants with different BMI categories at baseline. Adjusted for sex, ethnicity, and age (follow-up LVMi and percentage change in LVMi additionally adjusted for baseline LVMi). ^a,b,c^ Indicate significant difference between BMI categories, where data points or bars with the same superscript letter differ significantly (*p* < 0.05).

**Figure 2 nutrients-12-03185-f002:**
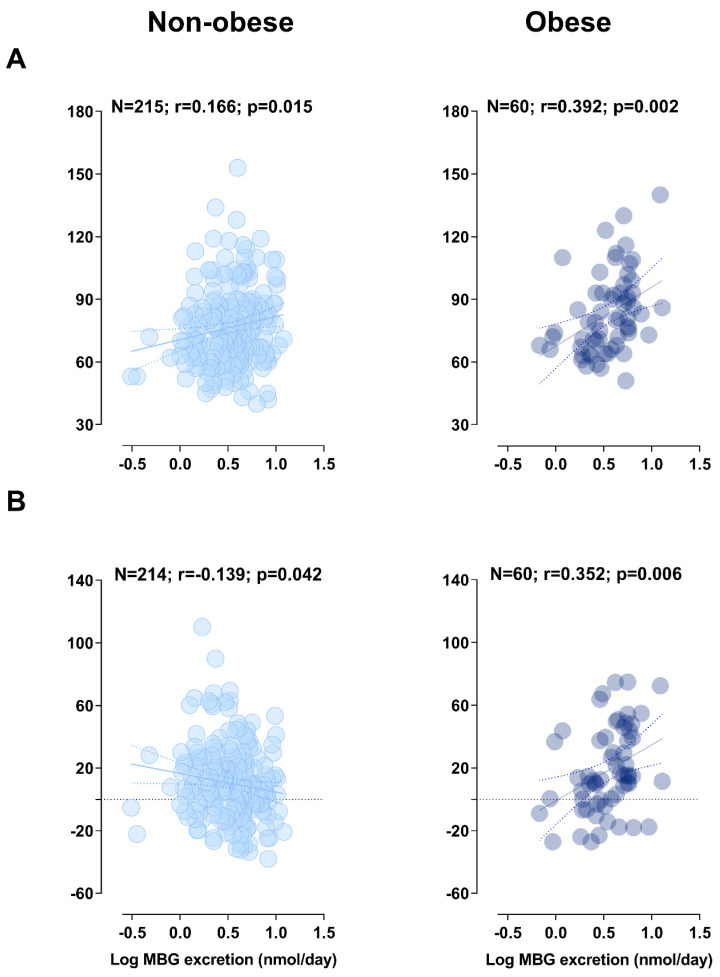
Pearson correlations of (**A**) follow-up LVMi and (**B**) the percentage change in LVMi with baseline MBG excretion in non-obese and obese participants.

**Table 1 nutrients-12-03185-t001:** Characteristics of 275 participants followed over 4.5 years.

	Baseline	Follow-Up	Difference	*p*
Men, *N* (%)	125 (45.5)	125 (45.5)		
Black, *N* (%)	138 (50.2)	138 (50.2)		
Age (years)	25.4 ± 3.16	30.0 ± 3.21	4.60 (4.49; 4.70)	<0.001
Anthropometric measurements				
Height (m)	1.68 ± 0.09	1.68 ± 0.09	0.00 (−0.001; 0.001)	0.68
Weight (kg)	73.3 ± 18.3	77.9 ± 20.2	4.64 (3.51; 5.77)	<0.001
Waist circumference (cm)	81.5 ± 14.2	83.2 ± 14.5	1.66 (0.91; 2.41)	<0.001
BMI (kg/m^2^)	25.8 ± 5.79	27.3 ± 6.57	1.51 (1.21; 1.82)	<0.001
WHtR	0.48 ± 0.08	0.49 ± 0.09	0.01 (0.01; 0.01)	<0.001
Frequency of obesity based on:				
BMI, *N* (%)	60 (21.8)	82 (29.8)	22 (36.7)	<0.001
WC, *N* (%)	95 (34.5)	109 (39.6)	14 (14.7)	0.014
WHtR, *N* (%)	94 (34.2)	103 (37.5)	9 (0.9)	0.11
Composite obesity criteria, *N* (%) *	55 (20.0)	74 (26.9)	19 (34.5)	<0.001
Blood pressure				
Clinic SBP (mmHg)	120 ± 12.4	116 ± 12.7	−3.76 (−4.97; −2.55)	<0.001
Clinic DBP (mmHg)	78.9 ± 8.07	79.3 ± 9.45	0.30 (−0.63; 1.24)	0.52
Central SBP (mmHg)	109 ± 9.48	110 ± 10.4	0.87 (−0.12; 1.86)	0.085
Hypertension, *N* (%) ^#^	39 (14.2)	42 (15.3)	3 (7)	0.76
Hypertension medication, *N* (%)	0 (0.0)	3 (1.0)		0.25
Echocardiography				
LVMi (g/m^2^)	70.7 ± 15.7	77.8 ± 18.7	7.02 (5.03; 9.00)	<0.001
IVSd (cm/m)	0.47 ± 0.10	0.53 ± 0.09	0.06 (0.05; 0.08)	<0.001
LVIDd (cm/m)	2.84 ± 0.25	2.78 ± 0.24	−0.07 (−0.09; −0.04)	<0.001
PWTd (cm/m)	0.50 ± 0.09	0.54 ± 0.01	0.05 (0.03; 0.06)	<0.001
EDVi (mL/m)	64.0 ± 13.7	60.3 ± 13.0	−3.70 (−4.94; −2.46)	<0.001
SVi (ml/m^2.04^)	25.1 ± 5.52	23.1 ± 4.96	−2.03 (−2.61; −1.45)	<0.001
Urinary profile				
eGFR (ml/min/1.73 m^2^)	111 ± 16.4	108 ± 16.6	−3.15 (−4.90; −1.39)	<0.001
24 h MBG excretion (nmol/day)	3.38 (1.12; 9.13)	-		
Estimated salt intake (g/day) ^ꝉ^	7.73 (2.80; 19.4)	7.13 (1.61; 24.4)	−0.22 (10.3)	0.47
Biochemical profile				
Glucose (mmol/L)	4.63 ± 0.76	4.08 ± 0.65	−0.55 (−0.66; −0.44)	<0.001
HDL-C (mmol/L)	1.34 ± 0.39	1.25 ± 0.34	−0.09 (−0.12; −0.05)	<0.001
LDL-C (mmol/L)	2.80 ± 0.92	2.66 ± 0.91	−0.14 (−0.22; −0.06)	0.001
C-reactive protein (mg/L)	1.04 (0.11; 9.38)	1.07 (0.15; 10.3)	0.01 (1.46)	0.56
γ-glutamyl transferase (U/L)	21.9 (8.74; 61.1)	21.4 (7.21; 63.5)	−0.48 (9.12)	0.46

Data presented as mean ± SD and geometric mean (5th and 95th percentiles). Difference from baseline to follow-up represented as mean (95% Confidence intervals (CI)) for normally distributed data and median (Inter quartile range (IQR)) for non-parametric data. * Obesity: BMI > 30 kg/m^2^ [23] and WC > 94 cm for white men; >81.2 cm for black men; >80 cm for white women and >81 cm for black women [24] and WHtR >0.5 [23] ^#^ Hypertension: Clinic SBP ≥ 140 mmHg and/or DBP ≥ 90 mmHg ^ꝉ^ Estimated salt intake based on 24 h sodium excretion. BMI, body mass index; DBP, diastolic blood pressure; EDVi; end diastolic volume index; eGFR: estimated glomerular filtration rate; HDL-C: high density lipoprotein cholesterol; IVSd: interventricular septum at end-diastole; LDL-C: low density lipoprotein cholesterol; LVIDd: LV internal diameter at end-diastole; LVMi, left ventricular mass index; MBG, marinobufagenin; PWTd: posterior wall thickness at end-diastole SBP, systolic blood pressure; SVi, stroke volume index; WC, waist circumference; WHtR, waist/height ratio.

**Table 2 nutrients-12-03185-t002:** Multiple regression analyses with follow-up LVMi and percentage change in LVMi as dependent variables and baseline MBG excretion as the main independent variable.

	MBG Excretion (nmol/Day)					
	Non-Obese 18.6–29.9 kg/m^2^ *N* = 211			Obese BMI > 30 kg/m^2^ *N* = 56		
Dependent Variable	Adj *R*^2^	Std. β	*p*	Adj R^2^	Std. β	*p*
LVMi (g/m^2^)	0.39	NS		0.35	0.311	0.007
% Δ LVMi	0.21	NS		0.4	0.336	0.003
	Sensitivity analysis additionally adjusted for estimated salt intake					
LVMi (g/m^2^)	0.39	NS		0.35	0.311	0.008
% Δ LVMi	0.21	NS		0.4	0.337	0.003
	Sensitivity analysis additionally adjusted for estradiol					
LVMi (g/m^2^)	0.39	NS		0.47	0.305	0.007
% Δ LVMi	0.21	NS		0.5	0.344	0.002

Adjusted for sex, ethnicity, age, clinic SBP, eGFR, glucose, HDL, c-reactive protein (CRP), gamma-glutamyl transferase (GGT), and baseline LVMi. NS refers to *p* > 0.05.

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
