# Peer review of "The Cardiotonic Steroid Marinobufagenin Is a Predictor of Increased Left Ventricular Mass in Obesity: The African-PREDICT Study"

_nutrients, 2020, doi:10.3390/nu12103185_

Round 1

Reviewer 1 Report

In this manuscript, Strauss-Kruger and colleagues describes the results of a follow-up analysis on the association between marinobufagenin excretion and left ventricular mass index in a cohort of young adults. The results of the study support previous work that reported a relationship between increased excretion of MBG and increased LVMi.

The article is well written, as well as supported by the methods used and results. Overall, the article has high quality and the authors should be commended for their work.

My comments:

1) Given that only part of patients were analyized (those with available follow-up data on MBG and LVMi), potential selection bias may have contributed to the results. Was the cohort analyized in this study significantly different from other patients of the main cohort, regarding principal baseline charateristics (i.e. age, sex, weight, comorbidities etc.)? This would be important to ensure the correctness of the analysis. The authors may want to briefly report about the cause of the exclusion of the others patients (i.e. obviously they did not have available data, but why? e.g. they withdraw consent/were lost at follow-up etc.).

2) Data presentation may be improved. In current form, figures are a bit unclear to me. Fig. 1 Panel a: I cannot understand the legend. It is reported that "a", "b" "C" indicate significant differences between BMI categories, but these are not reported in the image. Fig. 1 Panel B, what does this "bubble plot" means? Why these "bubbles" are of different measures? Fig 2, panel A and B for non-obese are barely readable (it is impossible to look at scatterplot for each subgroup, so that it may be pointless to report this scatterplots for these three categories - if you want to report these patients together, it may be better to merge as "non-obese").

3) Given that the aim of the study was to demonstrate a potential relationship between MBG and LMVi in obese patients compared to non-obese, and also given the small sample size of the study, I am not sure that analyzing separately normal weight and overweight patients may be the correct choice. Are the results different if normal and overweight patients were analyzed together? The authors may want to provide this analysis at least as a supplementary analysis, if they do not want to use that as a primary model. Also, the authors may want to comment on the results of the model adjusted for estradiol in overweight patients (which is significant).

Other comments:
- Row 69-70: please fix the sentence ("expression" is repeated two times).

Reviewer 2 Report

  1. The manuscript entitled "The endogenous Na+/K+-ATPase inhibitor, marinobufagenin, is a predictor of an increased left ventricular mass index over 4.5 years in obese, but not lean, young adults: The African-PREDICT study" is too long and misleading the readers, for that kind replace it with a short and attractive title.
  2. Avoid the use of " We, our" in your MS.
  3. The similarity factor of your MS is high (33%) kindly decrease it to be less than 20%.
  4. use the following paper which is very relative to your work" P3.12 A MONOCLONAL ANTIBODY TO THE ENDOGENOUS NA/K-ATPASE LIGAND, MARINOBUFAGENIN, REDUCES PROFIBROTIC GENE EXPRESSION AND REVERSES CARDIOVASCULAR FIBROSIS IN SALT-SENSITIVE HYPERTENSION".
  5. The MS needs language and editing correction.

Author Response

This manuscript is a resubmission of an earlier submission. The following is a list of the peer review reports and author responses from that submission.